# Identifying Active Rather than Total Methanotrophs Inhabiting Surface Soil Is Essential for the Microbial Prospection of Gas Reservoirs

**DOI:** 10.3390/microorganisms12020372

**Published:** 2024-02-11

**Authors:** Kewei Xu, Cheng Tao, Lei Gu, Xuying Zheng, Yuanyuan Ma, Zhengfei Yan, Yongge Sun, Yuanfeng Cai, Zhongjun Jia

**Affiliations:** 1State Key Laboratory of Shale Oil and Gas Enrichment Mechanisms and Effective Development, SINOPEC, Beijing 100083, China; taocheng.syky@sinopec.com (C.T.); gulei.syky@sinopec.com (L.G.); zhengxuying.syky@sinopec.com (X.Z.); mayuanyuan.syky@sinopec.com (Y.M.); 2SINOPEC Key Laboratory of Petroleum Accumulation Mechanisms, Wuxi 214126, China; 3Wuxi Research Institute of Petroleum Geology, Research Institute of Petroleum Exploration & Production, SINOPEC, Wuxi 214126, China; 4School of Biotechnology, Jiangnan University, Wuxi 214122, China; zhengfeiyan@jiangnan.edu.cn; 5Department of Earth Science, Zhejiang University, Hangzhou 310027, China; ygsun@zju.edu.cn; 6Institute of Soil Science, Chinese Academy of Sciences, Nanjing 210008, China; yfcai@issas.ac.cn; 7State Key Laboratory of Black Soils Conservation and Utilization, Chinese Academy of Sciences, Changchun 130102, China

**Keywords:** microbial prospecting of oil and gas (MPOG), methane-oxidizing bacteria, stable isotope probing, high-throughput sequencing, biotic index (BI), anomaly

## Abstract

Methane-oxidizing bacteria (MOB) have long been recognized as an important bioindicator for oil and gas exploration. However, due to their physiological and ecological diversity, the distribution of MOB in different habitats varies widely, making it challenging to authentically reflect the abundance of active MOB in the soil above oil and gas reservoirs using conventional methods. Here, we selected the Puguang gas field of the Sichuan Basin in Southwest China as a model system to study the ecological characteristics of methanotrophs using culture-independent molecular techniques. Initially, by comparing the abundance of the *pmoA* genes determined by quantitative PCR (qPCR), no significant difference was found between gas well and non-gas well soils, indicating that the abundance of total MOB may not necessarily reflect the distribution of the underlying gas reservoirs. ^13^C-DNA stable isotope probing (DNA-SIP) in combination with high-throughput sequencing (HTS) furthermore revealed that type II methanotrophic *Methylocystis* was the absolutely predominant active MOB in the non-gas-field soils, whereas the niche vacated by *Methylocystis* was gradually filled with type I RPC-2 (rice paddy cluster-2) and *Methylosarcina* in the surface soils of gas reservoirs after geoscale acclimation to trace- and continuous-methane supply. The sum of the relative abundance of RPC-2 and *Methylosarcina* was then used as specific biotic index (BI) in the Puguang gas field. A microbial anomaly distribution map based on the BI values showed that the anomalous zones were highly consistent with geological and geophysical data, and known drilling results. Therefore, the active but not total methanotrophs successfully reflected the microseepage intensity of the underlying active hydrocarbon system, and can be used as an essential quantitative index to determine the existence and distribution of reservoirs. Our results suggest that molecular microbial techniques are powerful tools for oil and gas prospecting.

## 1. Introduction

Oil and gas exploration serves as the fundamental cornerstone for the development and utilization of petroleum [1]. In recent years, the persistent deepening of oil and gas target layers coupled with the increasing complexity of reservoir types has presented substantial hurdles for traditional geophysical and geochemical techniques. Microbial prospection for oil and gas (MPOG), a rapidly advancing exploration technology, exhibits characteristics of low ambiguity, high efficiency, and environmental friendliness, garnering increasing attention from petroleum explorationists [2]. MPOG technology is based on the theory of hydrocarbon microseepage. Driven by the continuing pressure underground, volatile components originating from oil and gas reservoirs can vertically permeate the overlying strata and ascend to the Earth’s surface. These gaseous and volatile hydrocarbons have the potential to influence the distribution and proliferation of the hydrocarbonoclastic microorganisms in topsoils [3]. The discernment of the activity and distribution of these highly specialized populations serves as a predictive tool for anticipating the presence of oil and gas deposits [4].

The core of MPOG lies in the identification of oil and gas indicator microorganisms [5]. Common indicator microorganisms include C_1_–C_4_ gaseous hydrocarbon-oxidizing bacteria, in particular methane-oxidizing bacteria (MOB), which are the most extensively used indicators for oil and gas exploration. Over the past century, a comprehensive understanding of the physiological and biochemical characteristics of MOB has been achieved. Cultivated aerobic methanotrophs, including three families and over twenty genera, are divided into γ-Proteobacteria (type I and type X), α-Proteobacteria (type II and type III) and *Verrucomicrobia* (type IV) groups [6]. With the development of molecular ecological methods, recent studies have discovered uncultured microorganisms in soil that oxidize atmospheric methane, namely “USC-α and USC-γ” clusters [7], but pure strains have not yet been isolated. Whether these uncultured atmospheric methanotrophs can be used as indicators for oil and gas remains unknown to date.

Microbial methane oxidation is primarily catalyzed by methane monooxygenase. The *pmoA* gene, responsible for encoding the β-subunit of particulate methane monooxygenase (pMMO), stands out as one of the most valuable functional and phylogenetic biomarkers for methanotrophs. Its widespread utilization has been instrumental in exploring the diversity of methanotrophic communities in various environments [8]. In the past decade, qPCR-based analysis of microbial anomalies has been proposed and has successfully reflected the existence of underlying oil and gas reservoirs by comparing the abundance of *pmoA* genes in soils collected from oilfields, gas fields and non-oil and gas blocks, as well as different regions of oil and gas fields [9,10,11]. However, as a special functional microbial group, the abundance and activity of methanotrophs in various inhabits are significantly shaped by biotic and abiotic environmental factors, such as CH_4_ and O_2_ concentrations, pH level, temperature range, moisture content, nutrient availability and plant cover [6,12]. In our previous studies, *Methylobacter*-related groups were found to be indicator species above an onshore Chinese oil and gas reservoir, whereas the type II *Methylosinus* was indicative of the active population in the background soils. Similar results were observed in abandoned oil exploration wells [13] and terrestrial natural gas seeps [14]. However, conflicting results were recently reported by other petroleum explorers [15,16], who detected anomalies of type II methanotrophic *Methylocystis* and *Methylosinus* above typical oil reservoirs. Based on these inconsistencies, we thus hypothesize that active methanotrophs are more reliable indicators for oil and gas reservoirs than total methanotrophs.

With the rapid advancement in molecular ecological techniques, in particular the prominent “DNA-based stable isotope probing” (DNA-SIP) technology garnering significant attention in recent years, researchers now have the capability to employ stable isotopes for in situ tracing of oil- and gas-related microorganisms. Recently, the abundance and community composition of active gaseous-alkane degraders in various hydrocarbon macroseeps, such as natural gas seeps [17], marine hydrocarbon seeps [18], and freshwater lake sediments [19], have been effectively surveyed. However, knowledge of the ecological characteristics of active MOB at hydrocarbon microseeps is still lacking to date. Furthermore, MOB typically exhibit relatively low abundance in natural habitats, making it challenging to accurately characterize their structure and abundance using conventional culture approaches. High-throughput sequencing (HTS) technology is another powerful tool, capable of providing the highest resolution for low abundance microorganisms and revealing subtle changes in the composition and abundance of oil and gas indicator methanotrophs [20].

Here, we selected the Puguang gas field of the Sichuan Basin in Southwest China as a model survey area to (1) quantitate and contrast the *pmoA* gene abundance of total methanotrophic bacteria in non-gas field and gas field soils, (2) identify active methanotrophs by using a combination of ^13^C-DNA stable isotope probing and high-throughput sequencing, and (3) correlate the active MOB anomalies with the gas reservoir distribution to evaluate the effectiveness of molecular microbial techniques on MPOG.

## 2. Materials and Methods

### 2.1. Site Description

Soil samples were collected from the Puguang gas field (31.45°~31.65° N, 107.65°~107.90° E) located in the northeastern Sichuan fold-thrust belt within the Sichuan Basin, Southwest China (Figure 1). This gas field is a structural-stratigraphic trap, formed by the interplay of lateral depositional changes and fault closure. Its genesis can be traced back to a paleo-oil reservoir originating in the Triassic–Jurassic period. During profound burial in the Jurassic–Cretaceous, thermogenic gas was trapped from Lower-middle Silurian and Permian rocks. The paleotrap was subsequently metamorphosed into the present gas reservoir due to Tertiary–Quaternary compression [21]. The gas is primarily confined within the Lower-Triassic Feixianguan and the Upper-Permian Changxing reservoirs, which are predominantly composed of dolomitized oolites deposited in shelf and platform-margin shoal and backreef environments. The reservoir exhibits notable characteristics, featuring porosity ranging from 1% to 29% and permeability spanning 0.01 to 9664 mD, with burial depths exceeding 5000 m (16,400 ft) [22].

### 2.2. Sample Collection

A handheld GPS was used to position each sampling site with a measuring accuracy within five meters. As shown in Figure 1, to build a forward model, samples were firstly collected above earlier drilled and known areas (site P1~P3: gas wells, site J1~J3: dry wells). These samples embedding oil and gas information were selected for subsequent quantitative PCR, MiSeq sequencing and DNA-SIP analysis to identify active MOB. To investigate the biogeographical distribution of the relative abundance of active MOB, a total of 40 sampling sites were planned and collected from the northwest–southeast-direction survey line of the Puguang gas field for MiSeq sequencing of 16S rRNA and *pmoA* genes. In the survey area, the surface soil was classified as a purplish soil based on the FAO (Food and Agriculture Organization of the United Nations) system, with a sandy loam texture and pale purple color. The soil characteristics were as follows: water content, 10.8% (6.6–15.2%); total organic carbon, 1.34 g kg^−1^ (0.31–16.66 g kg^−1^); headspace CH_4_, 3.23 mL/L (2.18–17.55); total N, 0.72 g kg^−1^ (0.32–2.60); total P, 377.1 mg kg^−1^ (161–793 mg kg^−1^); and pH 6.8 (4.8–7.9). All the soil samples were collected in quintuplicate, with each sample weighing 200 g, from an optimal depth of approximately 60 cm, as previously described [3]. This depth was chosen to minimize anthropogenic disturbances. Subsequently, the samples were thoroughly blended, securely packed in sterile bags, and stored at temperatures of 4 °C and −20 °C for DNA-SIP incubation and DNA extraction, respectively.

### 2.3. DNA Extraction and Real-Time Quantitative PCR

The genomic DNA of soil microbes was extracted using the FastDNA Spin kit for soil (MP Biomedicals) according to the manufacturer’s instructions. To detect variations in the abundance of methanotrophic communities and evaluate the ^13^CH_4_ labeling of methanotrophs, the copy number of *pmoA* genes in both the total DNA extracts and DNA gradient fractions was determined by real-time quantitative PCR (qPCR) using a CFX96 Optical Real-Time detection system (Bio-Rad Laboratories). The primer pairs were A189f-mb661r [23] for *pmoA* genes and 515f-907r [24] for 16S rRNA genes. The qPCR procedures for both the *pmoA* genes and the 16S rRNA genes were identical except for the primers. Real-time qPCR analysis was performed in 20 µL volumes, including 10 µL 2 × SYBR qPCR Mix (Takara), 2 μL of each primer (5 μM), 400 ng DNA template, and ddH_2_O. The PCR protocol consisted of an initial step at 95 °C for 1 min, followed by 40 cycles of 95 °C for 30 s, 58 °C for 30 s, 72 °C for 30 s, and a final extension at 80 °C for 5 s. Standards were prepared using plasmid DNA from a representative clone containing *pmoA* or 16S rRNA genes, and a dilution series of the standard template ranging from 10^2^ to 10^9^ per assay was used. To ensure the specificity of amplification, melting curve analysis was performed at a heating rate of 0.5% from 60 to 96 °C at the end of each PCR run.

### 2.4. Stable Isotope Probing Microcosms

Stable isotope probing microcosms with CH_4_ were conducted in triplicate for gas well soil (P1) and dry-well soil (J3), respectively. Soil equivalent to 6.0 g dry weight was incubated in a 120 mL serum bottle sealed with a butyl rubber stopper at 28 °C in the dark, at approximately 60% of its maximum water-holding capacity. The CH_4_ concentration in the headspace was adjusted and maintained at 25,000 ppmv. ^13^CH_4_ and ^12^CH_4_ were performed as the labelled and control treatments, respectively. During the SIP incubation, the headspace methane concentration was measured by gas chromatography as previously described [25]. Soils were harvested at the desired time, pooled, and then stored at −20 °C for further study. Each microcosm incubation was terminated when approximately 80% of CH_4_ was consumed, i.e., when the CH_4_ concentration in the headspace dropped below 5000 ppmv, or after 6 weeks if the CH_4_ concentration in the headspace remained above 5000 ppmv. The soils were then collected and stored at −80 °C for subsequent DNA extraction.

### 2.5. Identification of ^13^C-Enriched DNA

As described above, a FastDNA spin kit for soil (MP Biomedicals) was used to extract total genomic DNA from 0.5 g of soil. After determining the quality and quantity of the DNA extracts using a NanoDrop ND-2000 UV–visible light spectrophotometer (NanoDrop Technologies), ^13^C-DNA was separated from ^12^C-DNA in the total DNA extracts by isopycnic density gradient centrifugation. Briefly, about 2.5 μg of the extracted DNA was mixed with cesium chloride (CsCl) solution to a final volume of 5.5 mL with a buoyant density of 1.725 g mL^−1^. The mixtures were then ultracentrifuged at 45,000 rpm and 20 °C for 44 h using a Vti65.2 vertical rotor (Beckman Coulter). The DNA fractions for each sample were then collected, and CsCl density was measured following established protocols [26,27]. After precipitation with polyethylene glycol 6000, the fractionated DNA was purified using 70% ethanol and dissolved in 30 μL of sterile water.

### 2.6. High-Throughput Sequencing of 16S rRNA and pmoA Genes

The compositions of methanotrophs in the soils were investigated using Illumina MiSeq sequencing. Universal primers targeting 16S rRNA and *pmoA* genes were used to analyze total microbial communities in all soil microcosms, providing insights into proportional changes in methanotrophs relative to the overall microbial communities in soils. Moreover, the DNA obtained from ‘heavy’ CsCl fractions 4–6, with a density of 1.740–1.748 g mL^−1^ in ^13^C-labeled incubations, was analyzed using amplicon-based sequencing targeting both the 16S rRNA and *pmoA* genes. Simultaneously, the ‘light’ DNA fractions 8–10, with a density of 1.719–1.728 g mL^−1^, retrieved from the ^12^C control incubations were subjected to 16S rRNA gene sequencing to dissect the methanotrophic community in the background area. The *pmoA* and 16S rRNA genes were amplified using the primer pairs A189f- mb661r [23] and 515f-907r [24], respectively. Each forward primer was fused with a unique barcode sequence. The PCR primers and conditions followed the previously described protocol [26]. The resulting PCR products were purified by gel electrophoresis and pooled in equimolar ratios in a single tube. Sequencing samples were prepared using the Illumina TruSeq DNA kit. The purified library was subjected to a series of dilutions, denaturation, and re-dilutions, and then mixed with PhiX in accordance with the library preparation protocols. The sequencing was conducted using the MiSeq system (Illumina,) with paired-end sequencing (2 × 300 bp).

### 2.7. Bioinformatic Analyses

The Quantitative Insights Into Microbial Ecology (QIIME) pipeline [20,26] was employed for the analysis of the raw sequence files. Following this, a comprehensive set of 3.1 × 10^6^ high-quality 16S rRNA as well as 7.8 × 10^5^ *pmoA* gene sequences were effectively retained. The Ribosomal Database Project (RDP) classifier was used to taxonomically classify the representative sequences of all 16S rRNA OTUs. The *pmoA* reads were analyzed using a naive classifier, specifically the mothur “classify.seq” command, which has been previously described [28]. To conduct the phylogenetic analysis, one representative from each lineage, specifically the dominant OTU of each lineage, was selected. A phylogenetic tree for *pmoA* genes was constructed using the neighbor-joining method in MEGA X, incorporating bootstrapping with 1000 replicates for robustness [29].

### 2.8. Statistical Analysis

The abundance of methanotrophic bacteria in non-gas field and gas field soils was compared using a one-way analysis of variance accompanied by Tukey’s post hoc test. Significant differences between the two groups were assessed using Student’s *t*-test. *P*-value less than 0.05 was considered statistically significant. All statistical analyses were conducted using STATISTICA 6.0 (StatSoft, Inc., Tulsa, OK, USA).

In order to assess the level of diversity within methanotrophic communities, we calculated the Shannon–Wiener (H) diversity indices using the relative abundance values as the following equation:(1)$$H=\sum Pi\left({log}_{2}Pi\right)$$
where P is defined as the ratio between the specific *pmoA* gene group and the sum of all groups.

Evenness (E) was calculated as:(2)$$E=H/H_{max}$$
where H_max_ = log_2_(S).

Richness (S): total number of *pmoA* gene groups in the methanotrophic community. 

To study the biogeographical distribution of active MOB anomaly, the sum of the relative abundances of RPC-2 and *Methylosarcina* for each soil was calculated as a specific biotic index (BI) in the Puguang gasfield. An anomaly distribution map of BI values was generated using Surfer software 16.0 [30].

## 3. Results and Discussion

### 3.1. Analysis of pmoA Gene Abundance and Composition above Gas and Non-Gas Fields

Real-time quantitative PCR was employed to accurately quantify the abundance of total bacteria (16S rRNA) and methane-oxidizing bacteria (*pmoA*) in soil samples collected from gas wells (P1, P2, P3, P4) and dry wells (J1, J2, J3). As shown in Figure 2A, the abundance of 16S rRNA genes varied from 10^9^ to 10^10^ copies/g, with a tenfold difference between the maximum and minimum values. The methanotrophic *pmoA* gene contents ranged from 10^7^ to 10^8^ copies/g, with an average value of 6.03 × 10^7^ copies/g in gas well soils and 6.24 × 10^7^ copies/g in dry-well soils, even slightly higher in the case of dry-well soils. To minimize the impact of environmental disturbances on the soil samples, we thereafter normalized the abundance of *pmoA* genes to the total abundance of 16S rRNA genes. Surprisingly, no statistically significant difference (*p* = 0.087) was observed between gas field and non-gas field soils, with respect to the ratio of *pmoA*/16S rRNA gene copy numbers (Figure 2B). According to previous studies, quantitative PCR assay-based methanotrophic *pmoA* gene anomalies can effectively indicate the change in the horizontal distribution characteristics of underground oil and gas reservoirs [10,15]. The current study does not appear to support these earlier findings. 

Therefore, we further determined the composition of methanotrophic bacteria in gas well soil (P1) and dry-well soil (J3) by using MiSeq sequencing. In total, 18,000 high-quality *pmoA* sequences were obtained for each sample to compare with known functional gene databases [28]. The sequencing results showed that the *pmoA* sequences were classified into 33 known genotypes, while others could only be identified as type I or type II MOB. Figure 2C lists ten dominant *pmoA* genotypes in the Puguang gas field. The gas field soil (P1) was mainly dominated by *Methylocystis* and RPC-2 (rice paddy cluster-2) [31], with percentages of 40.0% and 25.6%, respectively. The sub-dominant species were *Methylosarcina* and RPCs with 14.6% and 11.0%, respectively. In contrast, the composition of methane-oxidizing bacteria in dry-well soil (J3) was heterogeneous, with USC-γ as the only dominant taxon, accounting for almost 90% of the total MOB. This absolute dominance is similar to that previously reported in the same Qinling region [32,33]. USC-γ is a *pmoA* genotype that is widely distributed in dryland soils [34] and is thought to be closely associated with the oxidation of low atmospheric concentrations of methane [35,36]. Pure cultivated strains of this genotype have not yet been obtained.

Based on the above results, it can be speculated that the abundance of total methanotrophs may not necessarily reflect the distribution of the underlying gas reservoirs. The significant differences in methanotrophic communities between gas field and non-gas field soils indicate that different species have different indicative effects on the presence of natural gas. It is therefore essential to accurately identify the active MOB in order to improve exploration accuracy.

### 3.2. Stable Isotope Probing (SIP) of Active Methanotrophs in Gas Field Soils

To investigate the methanotrophs that were active in assimilating methane in the soils collected from a gas field and non-gas field, DNA-SIP incubations were carried out using ^13^CH_4_. As shown in Figure 3A, gas well soil P1 exhibited rapid methane oxidation after 13 days, with almost all methane oxidation completed by day 44, resulting in a methane oxidation percentage of 99%. On the other hand, dry-well soil J3 showed a weaker oxidation capacity compared to P1. It entered the stage of rapid methane oxidation after 26 days of cultivation. Even after 57 days, the methane concentration in the bottle remained at approximately 3000 ppmv, with a methane oxidation percentage of 87%. These two samples were subsequently selected for DNA-SIP and high-throughput sequencing analysis.

Buoyant density centrifugation was performed with the corresponding DNA extracts from ^12^CH_4_ and ^13^CH_4_ microcosms. The quantitative distribution of *pmoA* genes in these gradients was analyzed by qPCR of DNA using primer pairs for total methanotrophs. At the end of SIP incubations, the DNA concentration profiles from 14 different density fractions displayed a clear separation of ‘heavy’ (^13^C) DNA from ‘light’ (^12^C) DNA (Figure 3B). Briefly, the peak of heavy DNA was observed at 1.74 to 1.75 g mL^−1^ (fractions 4–6) density fraction, whereas that of light DNA was observed at 1.72 to 1.73 g mL^−1^ (fractions 8–10), indicating successful incorporation of ^13^C into the microbial DNA involved in CH_4_ oxidation. Electrophoresis of PCR products of the *pmoA* gene in different buoyancy density fractions also clearly indicated the migration of methanotrophic bacteria DNA from ^13^CH_4_ labelled samples to the heavy fraction (Figure 3C). This result is in accordance with earlier SIP studies conducted to elucidate active MOB in similar environments, i.e., oil sands process water and mangrove sediments contaminated by oil spills [37,38].

Combining the above results, it was determined that fractions 4, 5 and 6 were ‘heavy’ ^13^C-labelled DNA, and the methanotrophic species in the DNA were considered to be highly active gas indicator bacteria. To identify these bacteria, MiSeq sequencing technology was used at the level of 16S rRNA genes and *pmoA* functional genes. The phylogenetic relationships of these indicator bacteria were further analyzed.

### 3.3. Identification of Active Methanotrophs Using MiSeq Sequencing of 16S rRNA Genes

All DNA fractions were amplified with the universal primers for 16S rRNA genes, and the products were mixed equimolarly and analyzed by MiSeq sequencing, yielding a total of 3.17 million high-quality DNA sequences, with an average of approximately 80,000 sequences per fraction. After comparative analyses with the 16S rRNA ribosomal taxonomic database, we obtained the taxonomic information at genus level. As shown in Figure 4, in agreement with the qPCR results of the *pmoA* gene (Figure 3B), ^13^C-labelled MOB in gas well soil P1 were mainly distributed in ‘heavy’ DNA fractions 4–6, which accounted for 38.73%, 32.41% and 20.26% of the total microorganisms, respectively (Figure 4A). The labelling fraction was relatively low compared to paddy soils (usually above 60%) [27]. This could be attributed to the indirect acquisition of labelling by other non-MOB through the uptake of ^13^C-containing metabolites of MOB (cross-feeding) due to longer incubation time [39,40,41]. In contrast, the percentage of MOB in these three fractions corresponding to the ^12^CH_4_-treated control was less than 1%, suggesting that methanotrophic species in the ^13^C-labelled ‘heavy’ DNA fractions 4–6 were the main contributors to methane oxidation. The distribution of MOB in each of the DNA fractions of the dry-well soil J3 was similar to P1, except that the percentages of ^13^C-labelled MOB in fractions 4–6 were much lower, only 13.31%, 8.75% and 5.49%, respectively (Figure 4B). This may be due to the fact that J3 was incubated for a longer period of time than P1 (nearly 2 months) (Figure 3A) and cross-feeding occurred more frequently. Nevertheless, this percentage is still much higher than that of the corresponding three fractions treated with ^12^C-methane (<0.5%).

In the gas well soil P1, phylogenetic analysis of ^13^C-labelled 16S rRNA genes from ‘heavy’ DNA fractions indicated that type II *Methylocystis* dominated the ^13^C-labelled methanotrophs (up to 33.76%), followed by *Methylosarcina* (2.23%), *Methylocadum* (2.20%), and *Methylmicrobium* (0.14%) (Figure 4A). In addition, two type I (*Methylobacter*, *Methylococcus*) and two type II (*Methylosinus*, *Methylocella*) methanotrophs were also detected, but their abundance in both the ^13^CH_4_ and ^12^CH_4_ incubations were extremely low and not indicative of methane oxidation. In situ composition of these genera was also consistent with the above results (Appendix A). *Methylocystis* accounted for 0.018%, which is less than the 0.078% of *Methylosarcina* and comparable to the 0.016% of *Methylobacter*. However, the composition of MOB completely changed after the SIP incubations. The relative abundance of *Methylocystis* was much higher than that of *Methylosarcina*, while the percentage of *Methylobacter* remained at its original low level. Therefore, based on the analysis of ^13^C-based high-throughput probing of 16S rRNA, it can be concluded that *Methylocystis* and *Methylosarcina* are the highly active MOB above gas reservoirs.

In the dry-well soil J3, five species of MOB were detected in ‘heavy’ DNA fractions, but were significantly different from P1. Type II *Methylocystis*, accounted for up to 99.6% of all labelled MOB, whereas the remaining species were detected at extremely low levels in both ^13^CH_4_ and ^12^CH_4_ incubations. Moreover, based on the sequencing of in situ soil sample, only three MOB sequences (*Methylococcus*, *Methylosoma* and *Methylocella*) were detected out of a total of 75,191 sequences, while *Methylocystis* was not detected. Therefore, combining the in situ and DNA-SIP results, it can be assumed that *Methylocystis* was the absolutely predominant active MOB in the background zone of the Puguang gas field.

### 3.4. Identification of Active Methanotrophs Using MiSeq Sequencing of pmoA Genes

For more detailed identification of gas indicator bacteria, the ‘heavy’ DNA (fractions 4–6) were amplified with functional gene *pmoA* universal amplification primers and also analyzed by MiSeq high-throughput sequencing. The DNA sequences were used to resolve the species and the relative abundance of individual methanotrophic ecotypes, and to deeply analyze oil and gas indicator bacteria at the functional gene level. A total of 269,000 high-quality sequences were obtained from the ‘heavy’ DNA fractions of soil P1 and J3, averaging about 45,000 sequences per fraction. By comparison with the *pmoA* functional gene database, a total of 32 different *pmoA* genotypes were detected, affiliating to multiple clusters such as type I, type II, pmoA2 and pxmA, including known culturable species, and a few ecotypes for which representative pure-culture strains are not yet available. Table 1 lists the top 10 abundant *pmoA* genotypes in the DNA-SIP ‘heavy’ fractions.

As shown in Table 1, the *pmoA* gene sequencing results were basically consistent with the 16S rRNA gene. The ‘heavy’ DNA fractions of soil P1 were also dominated by type II *Methylocystis*, especially the fourth fraction, accounting for almost 90% of the total MOB, followed by RPC-2 (rice paddy cluster-2), mainly distributed in the fifth and sixth fractions, accounting for about half of the MOB. RPC-2 was originally discovered in rice field soil [31], and no pure-culture strain has yet been isolated. Therefore, this unculturable methanotroph was not detected by MiSeq sequencing of the 16S rRNA. According to phylogenetic analysis (Figure 5), it is affiliated to a type Ia methanotroph and has a close genetic relationship with *Methylosarcina*. The relative abundance of *Methylosarcina* ranked third, accounting for 19.43% in the 6th fraction, which was comparable to 16.45% by 16S rRNA gene sequencing analysis.

The remaining genotypes accounted for much smaller percentages and may have only a relatively weak indication of methane oxidation, such as *pmoA*-2, which encodes the methane monooxygenase pMMO2, an isoenzyme of the traditional methane monooxygenase pMMO [42], responsible for maintaining the activity of *Methylocystis* and *Methylosinus* in stressful environments with low methane concentrations [23,43]. As can be seen from Table 1, the percentage of *Methylosinus* is relatively low, so the *pmoA*-2 genes detected in this study should mainly originate from *Methylocystis*. RPCs (rice paddy clusters) is another genotype originally found in rice paddies, which belongs to type Ib MOB, and is more closely related to *Methylocaldum* (Figure 5). The percentage of RPCs in the heavy fractions is about 0.5–2%, and its gas indicating effect might be slightly weaker. *Methylocaldum* and *Methylobacter*, which are rarely detected in 16S rRNA gene sequencing, also have an extremely low percentage in the *pmoA* sequencing results, showing the consistency of these two sequencing approaches. It is noteworthy that small amounts of USC-γ and USC-α were also detected in the ‘heavy’ DNA fractions. These two genotypes, which are phylogenetically more closely related to type I and type II MOB, respectively, are widely distributed in various dryland soils [34,44], and are considered to be the major players in atmospheric methane oxidation [35,36]. They therefore have a high affinity for low concentrations of methane, but are often unable to compete with culturable methanotrophs under conditions of high methane concentrations [45].

In contrast, the *pmoA* sequencing results of the ‘heavy’ DNA fractions of the dry-well soil (J3) are consistent with those of the 16S rRNA genes. *Methylocystis* accounts for as much as 88.72~97.44% of the total MOB. The percentage of RPC-2 in second place is much lower than that of *Methylocystis*, accounting for only about 5.93% in the sixth fraction. Moreover, all other species have very low relative abundances and may not be indicative of methane oxidation. 

In terms of ecological indices, the unbalanced distribution of the methanotrophic community in the dry-well soil (J3) led to a low Shannon’s diversity index (H = 0.53) and species evenness (E = 0.15). On the contrary, however, after geoscale acclimation to trace- and continuous-methane supply, the active type II *Methylocystis* in the gas well soil (P1) became less abundant, and the niche vacated by these dominant methanotrophs was gradually filled with type I RPC-2 and *Methylosarcina*. The replacement of type II MOB also resulted in significantly higher Shannon’s diversity and species evenness indices (H = 1.68; E = 0.48).

The above findings are consistent with our previous study. The type I *Methylobacter*-related group were found to be indicator species above a typical onshore oil and gas reservoir, whereas type II methanotrophic *Methylosinus* was indicative of the active population in the background soils [3]. Similar results have been observed in abandoned oil exploration wells [13] and terrestrial natural gas seeps [14]. However, the opposite results have also been reported by other petroleum explorationists [15,16], who detected anomalies of type II *Methylocystis* and *Methylosinus* in Chinese oilfields. The most likely reason is that the diversity of methanotrophs was influenced by abiotic and biotic environmental factors, such as CH_4_ and O_2_ concentrations, pH level, temperature range, moisture content and plant cover [46]. Therefore, stable isotope probing coupled with high-throughput sequencing is a powerful tool for identifying specific bioindicators associated with particular oil and gas fields, thereby improving the precision of subsequent exploration.

### 3.5. Correlation of the Active MOB Anomalies with the Distribution of Gas Reservoirs

The Puguang gas field is characterized as a large-structure–lithology composite gas pool, encompassing a confirmed gas enclosure area of approximately 50 km^2^. The primary gas payzones consist of dolomite reservoirs found in the Lower-Triassic Feixianguan Formation and the Upper-Permian Changxing Formation, which are prominently developed along the platform edge. These reservoirs, comprising oolitic dolomite in the Feixianguan Formation and biohermal limestone in the Changxing Formation, exhibit lateral overlap and vertical connectivity [22]. The current burial depths of the gas reservoirs in the Feixianguan Formation are relatively deep, averaging around 5200 m. In the Puguang gas field, six wells (P1, P2, P3, P4, P6, P9) have been drilled, and they have demonstrated high flow rates, from approximately (50~120) × 10^5^ m^3^/d before acidification. These results confirm the presence of thick, high-quality gas reservoirs. The natural gases extracted from these reservoirs are characterized as typically dry, containing minimal ethane and virtually no propane and higher alkanes [21]. In this study, the distribution of microbial anomalies determined by the combination of DNA-SIP and high-throughput sequencing was consistent with geological, geophysical data, and known drilling results.

Based on the aforementioned comparison of methanotrophic communities above dry and gas wells, we deduced that the active MOB in the Puguang gas field were RPC-2 and *Methylosarcina* (Figure 5). The sum of the relative abundance of these two methanotrophic species in the MiSeq sequencing data of the *pmoA* genes of each sample was used as the specific biotic index (BI) in the Puguang gas field to estimate the microbial anomaly at the different sites. The biogeographical distribution of BI values of soil samples collected from the northwest–southeast survey line is shown in Figure 6. In the gas field area, the BI values were significantly higher than in all other soil samples, ranging from 10.4% to 60.9%, with a mean of 24.9%. By contrast, in the non-gas-field area, the BI values were 8.9-fold lower, in the range from 0.1% to 6.9%. The boundary of the high-value anomaly is controlled by the Dongyuezhai–Puguang fault and the gas–water contact of the Puguang gas field, consistent with the areas of the gas accumulation (Figure 6). Therefore, the BI of active MOB successfully reflected the microseepage intensity of the underlying active hydrocarbon system, and can be used as an essential quantitative index to determine the existence and distribution of reservoirs.

It is worthy of highlighting that ^13^C-based high-throughput probing of active methanotrophs inhabiting surface soils is not only applicable to gas field exploration, but would be also quite effective in indicating oil reservoirs. In addition to the *pmoA* gene, other biomarkers of C_2_+ hydrocarbon-oxidizing bacteria, such as propane monooxygenase (*prmA*) [47], butane monooxygenase (*bmoX*) [48], and alkane hydroxylase gene (*alkB*) [3], can be integrated to accurately diagnose the full spectrum of microbial anomalies above oil and gas reservoirs.

## 4. Conclusions

This study aimed to test a hypothesis that active methanotrophs are more reliable indicators for oil and gas reservoirs than total methanotrophs. We selected the Puguang gas field in Southwest China as a model system to identify active methanotrophs by using a combination of ^13^C-DNA stable isotope probing and high-throughput sequencing, and then to correlate the active MOB anomalies with the gas reservoir distribution. It was found that type II methanotrophic *Methylocystis* was the absolutely predominant active MOB in the non-gas-field soils, whereas type I RPC-2 (rice paddy cluster-2) and *Methylosarcina* became more active in the surface soils of gas reservoirs after geoscale acclimation to trace- and continuous-methane supply. The sum of the relative abundance of RPC-2 and *Methylosarcina* was then used as specific biotic index (BI) in the Puguang gas field. The microbial anomaly distribution map based on the BI values showed that the anomalous zones were highly consistent with geological and geophysical data, and known drilling results. Our findings reinforce the significance of molecular microbial techniques on oil and gas prospecting. Further investigation into other biomarkers of C_2_+ hydrocarbon-oxidizing bacteria and their effectiveness is needed to accurately diagnose the full spectrum of microbial anomalies above oil and gas reservoirs. 

## Figures and Tables

**Figure 1 microorganisms-12-00372-f001:**
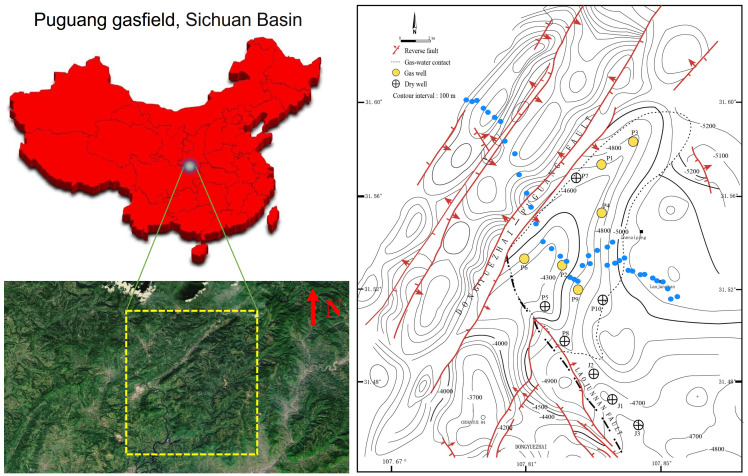
Geological setting of the Puguang gas field and sampling scheme for spatial analysis of the methanotrophic community. Yellow dotted box indicates survey area in the satellite image. Small blue circles indicate the soil samples collected from the northwest–southeast direction survey line for MiSeq sequencing of *pmoA* and 16S rRNA genes. Due to the difficulty of sampling the surface of the mountainous terrain, the sampling points are not necessarily arranged in a straight line.

**Figure 2 microorganisms-12-00372-f002:**
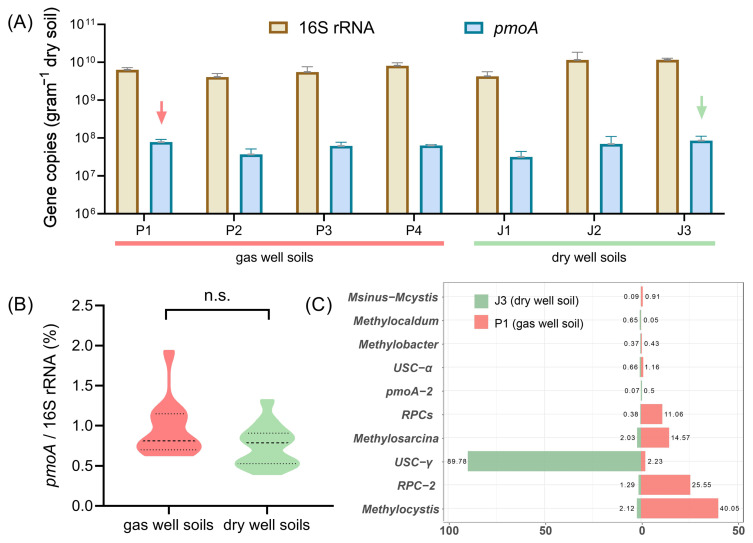
(**A**) Comparison of 16S rRNA and *pmoA* gene content in gas well soils (P1~P4) and dry well soils (J1~J3). Arrows indicate the soil samples for MiSeq sequencing of *pmoA* genes. (**B**) Comparison of the ratio of *pmoA*/16S rRNA gene copy numbers in non-gas field and gas field soils. (**C**) Compositions of methanotrophic bacteria in gas well soil (P1) and dry-well soil (J3) at the genus level.

**Figure 3 microorganisms-12-00372-f003:**
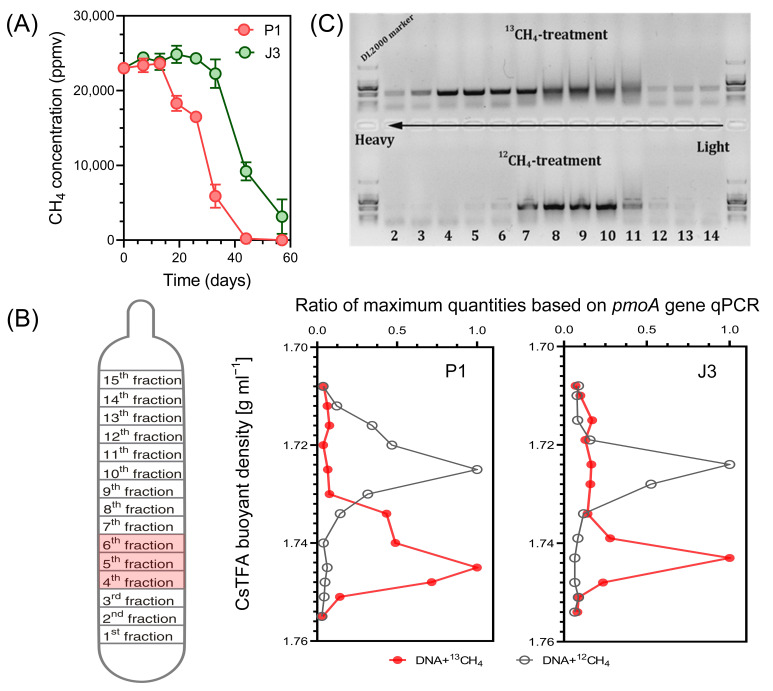
(**A**) Consumption of methane in the headspace of the soil P1 and J3 microcosms (n = 3). The incubation was terminated when approximately 80% of CH_4_ was consumed, or after 6 weeks if the CH_4_ concentration in the headspace remained above 5000 ppmv. (**B**) Distribution of ^13^C-labelled methanotrophs based on qPCR of *pmoA*. The ^13^C-labelled methanotrophs were identified by quantitatively analyzing the distribution of *pmoA* genes across the entire buoyant density gradient of DNA fractions collected from soil microcosms incubated with ^13^CH_4_, as compared to the control. The data has been normalized to represent the ratio of gene abundance in each DNA gradient to the maximum number for each treatment. (**C**) Electrophoresis of PCR products of the *pmoA* gene in different buoyancy density fractions.

**Figure 4 microorganisms-12-00372-f004:**
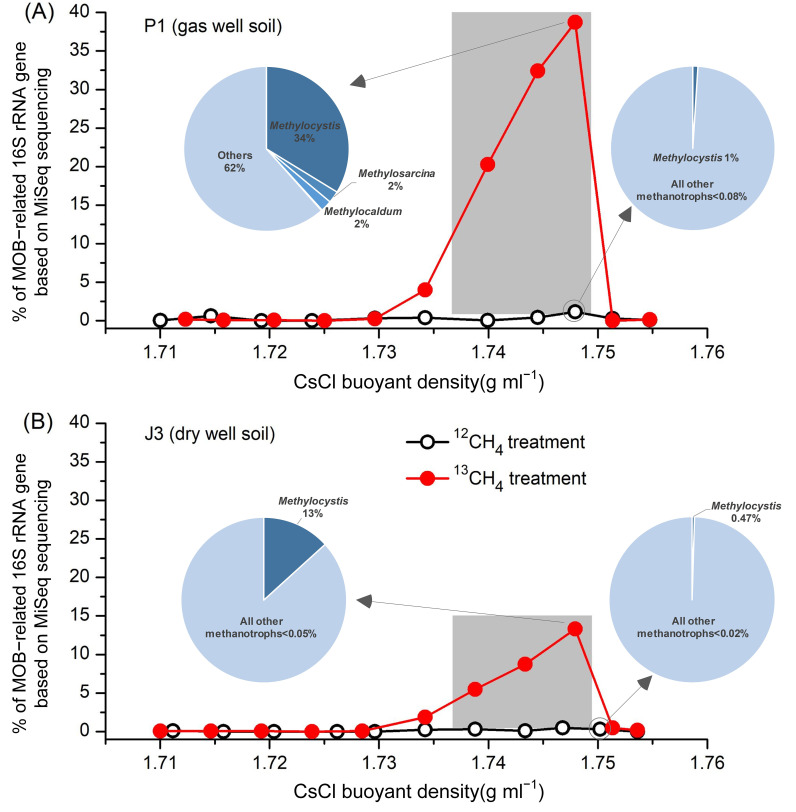
Percentage of MOB-related 16S rRNA gene based on MiSeq sequencing in each DNA fraction (fractions 2–12) of (**A**) gas well soil P1 and (**B**) dry well soil J3 from the Puguang gas field. Only methanotrophic groups are shown in the pie charts. Light-blue pie sections represent all other bacteria.

**Figure 5 microorganisms-12-00372-f005:**
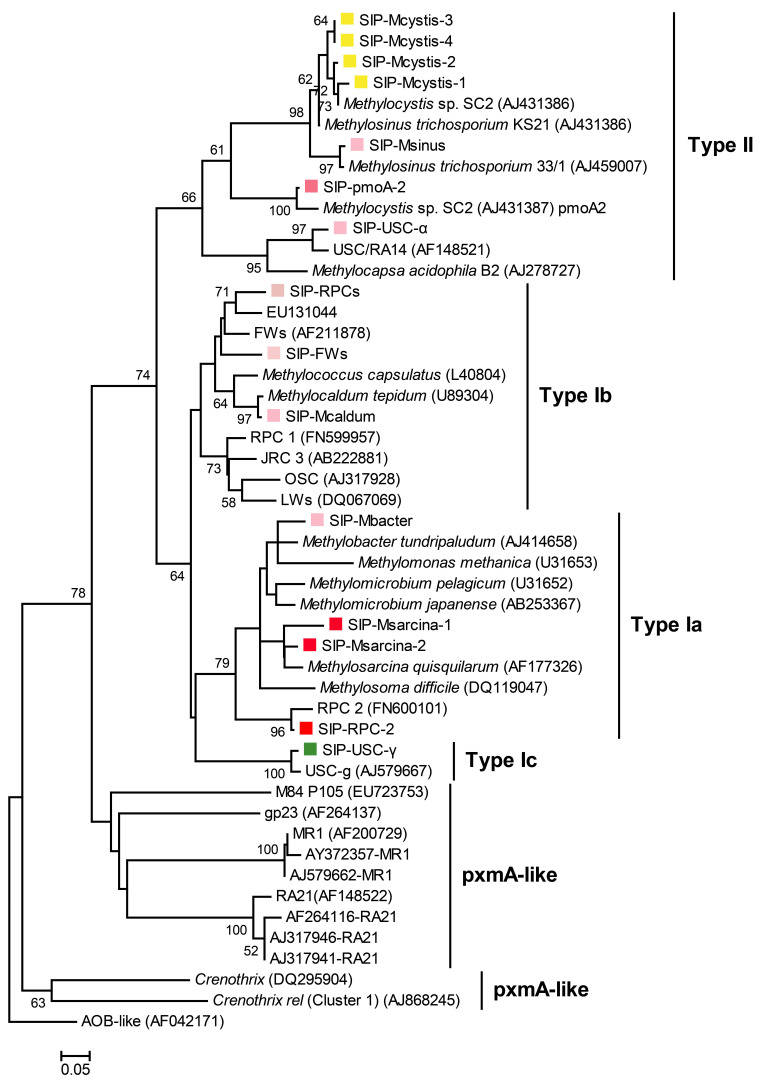
Phylogenetic neighbor-joining tree of *pmoA* gene sequences from ‘heavy’ DNA fractions constructed using MEGA X. Reference strains and clone sequences were retrieved from GenBank and are identified by their accession numbers. The scale bar represents 5% sequence divergence, while the values at the nodes indicate the percentages of 1000 bootstrap replicates supporting the branching order. Bootstrap values below 50% are not shown. Color square in front of sequences: shades of red in the square indicate the strength of the gas-field indication; yellow squares represent the shared genotype in both non-gas field and gas field soils; the green square indicates the dominant taxon in dry well soils in situ.

**Figure 6 microorganisms-12-00372-f006:**
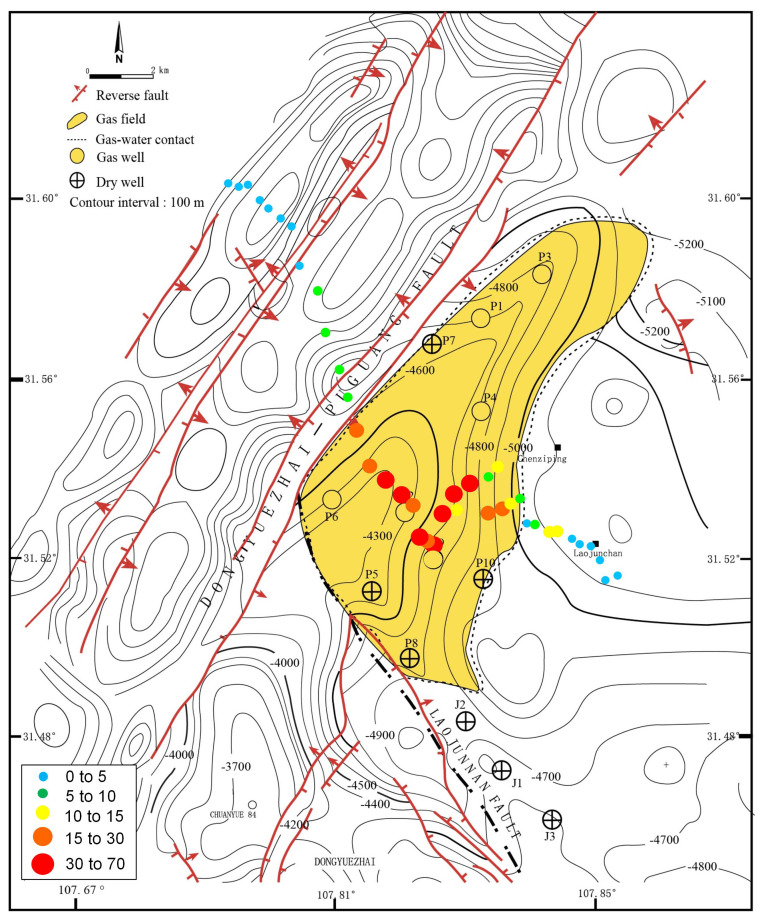
The biogeographical distribution of the biotic index (BI) values of soil samples collected from the northwest–southeast survey line of the Puguang gas field.

**Table 1 microorganisms-12-00372-t001:** Abundant methanotrophic genotypes and percentages (based on MiSeq sequencing of *pmoA* genes) in the DNA-SIP ‘heavy’ fractions of soil P1and J3.

*pmoA* Gene Group		Gas Well Soil (P1)			Dry Well Soil (J3)		
		4	5	6	4	5	6
type IIa	*Methylocystis*	89.49	45.01	26.93	91.69	97.44	88.72
type Ia	RPC-2	2.34	41.15	50.4	1.4	0	5.93
type Ic	USC-γ	0.89	0.67	0.61	1.94	0	1.48
type Ia	*Methylosarcina*	0.84	5.21	19.43	0.85	2.56	0
type Ib	RPCs	1.76	1.95	0.47	0.65	0	2.08
type IIb	pmoA-2	1.97	3.64	1.51	0.27	0	0.59
type IIb	USC-α	0.04	0.15	0.04	0.58	0	0
type Ia	*Methylobacter*	0.59	0.06	0.01	0	0	0.3
type Ib	*Methylocaldum*	0.01	0.01	0	0.1	0	0
type IIa	*Methylosinus*	0.05	0.02	0.01	0.78	0	0.3

## Data Availability

The raw amplicon sequence datasets for 16S rRNA and *pmoA* genes have been deposited in the National Center for Biotechnology Information Sequence Read Archive (SRA) under the accession numbers PRJNA1049154, PRJNA1049980 and PRJNA1049988.

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
