# Peer review of "Identifying Active Rather than Total Methanotrophs Inhabiting Surface Soil Is Essential for the Microbial Prospection of Gas Reservoirs"

_microorganisms, 2024, doi:10.3390/microorganisms12020372_

Round 1

Reviewer 1 Report

Comments and Suggestions for Authors

In this study, the authors evaluate the distribution of active MOBs in the Puguang gas field in the Sichuan Basin of southwest China using a combined method of 13C-DNA stable isotope exploration and high-throughput sequencing. The results reveal a correlation with gas reservoir distribution and demonstrate its potential as an important quantitative indicator for identifying the presence and distribution of reservoirs.

The methods used in this study are valid and well described. The research results are extremely interesting, as they quantitatively and appropriately demonstrate the correlation between the distribution of active methanotrophs in oil and gas storage areas and the oil and gas storage areas using this distribution variation as a bioindex. .

Questions are listed below as minor comments. We recommend that this paper be accepted after considering corrections on these points.

Minor comments:

I think it would be easier to understand the meaning of "active methylotroph" used in this paper, including the validity of the research method chosen for this study, so please consider adding this. Also, is there any intention behind using the word methylotroph, which has a broader meaning than methanotroph? The target bacteria of this research are methanotrophs, so wouldn't "active methanotrophs" be sufficient? Please consider making corrections or comments on this point as well.

Reviewer 2 Report

Comments and Suggestions for Authors

This interesting work, combining molecular and biomarker techniques, allows the recognition of hydrocarbon resources in crude oil and natural gas. The study areas were located in southwestern China. In my opinion that fully performed analysis of methanogens and methanotrophs, inhabiting surface soil,  seems to be more prospection.     

Reviewer 3 Report

Comments and Suggestions for Authors

The manuscript by Xu et al. describes the result of a study with the underlying hypothesis that active methylotrophs can be indicator strains for oil and gas reservoirs. The authors use stable-isotope probing, quantitative PCR, and high-throughput amplicon sequencing to conclude that methylotrophs of type Ia, specifically RPC-2 and Methylosarcina, were indicative for the Puguang gasfield.

This is a well written manuscript with a nice step-by-step logical flow.

There are just some minor details:

Lines 160-161: It does not make sense to give the volume in microliters for the PCR. For other researchers to repeat this reaction the concentrations and not only the volume of the primers, template and polymerase have to be indicated.

Line 383: “Table 1 lists …. for more than 1% …. : Table 1 lists many genera that constitute less than 1% in each fraction. Please clarify.

Line 392-393: I do not understand the logic of this sentence. I thought if you do MiSeq sequencing of 16S rRNA gene of a soil sample you will be able to detect many unculturable genera.

Line 404: The yellow squares are green on my screen and on my printout, so this leads to confusion, because green indicates different taxa.

Lines 433-437: Please clarify these sentences. “On the contrary” does not make sense to me. Contrary to what? What are the authors referring to with these two sentences?

Line 475: replace “was” with “is”
